# Solubility of Anthraquinone Derivatives in Supercritical Carbon Dioxide: New Correlations

**DOI:** 10.3390/molecules26020460

**Published:** 2021-01-17

**Authors:** Ratna Surya Alwi, Chandrasekhar Garlapati, Kazuhiro Tamura

**Affiliations:** 1Department of Chemical Engineering, Fajar University, Makassar 90231, Indonesia; 2Department of Chemical Engineering, Puducherry Technological University (Formerly Known as Pondicherry Engineering College), Puducherry 605014, India; chandrasekar@pec.edu; 3Division of Natural System, Graduate School of Natural Science and Technology, Kanazawa University, Kakuma-machi, Kanazawa 920-1192, Japan; tamura@se.kanazawa-u.ac.jp

**Keywords:** solubility, supercritical carbon dioxide, anthraquinone, AIC, new correlations

## Abstract

Solubility of several anthraquinone derivatives in supercritical carbon dioxide was readily available in the literature, but correcting ability of the existing models was poor. Therefore, in this work, two new models have been developed for better correlation based on solid–liquid phase equilibria. The new model has five adjustable parameters correlating the solubility isotherms as a function of temperature. The accuracy of the proposed models was evaluated by correlating 25 binary systems. The proposed models observed provide the best overall correlations. The overall deviation between the experimental and the correlated results was less than 11.46% in averaged absolute relative deviation (AARD). Moreover, exiting solubility models were also evaluated for all the compounds for the comparison purpose.

## 1. Introduction

Supercritical fluid (SCF) applications in process industry have gained a lot of momentum. The proper application solely depends on exact information on solubility, therefore, the estimation of solubility of a variety of substances in supercritical fluids has taking place in recent literature [1]. Among various supercritical fluids, carbon dioxide is one that has more attention due to its interesting and easily attainable critical properties [1]. Dyeing industry and pharmaceutical industry require solubility data, but the data are limited and available at particular specified temperatures and pressures [1]. Measuring solubility at each and every point would be a tedious task, therefore modeling is a must [2]. The present study is concerned about the modeling of anthraquinone derivatives in supercritical carbon dioxide. Anthraquinone derivatives are majorly used in dyeing industries—the exact prediction of solubility data are very much essential for the development of supercritical dyeing process. There are five frameworks through which solubility data are analyzed [3]. Out of five approaches, thermodynamic frameworks based on solid–gas equilibrium criteria and solid–liquid equilibrium criteria are very successful [4]. Solid–gas equilibrium approach requires critical, chemical, and physical information for the modeling. The availability of such information is very rare; therefore, solid–gas equilibrium approach entirely depends on group contribution methods for those necessary properties. Therefore, the solid–gas equilibrium approach would be purely lies on the accuracy of the predicted properties. Sometimes these properties may not be real and the corresponding correlation may not be appropriate. Therefore, we need to look for an alternative correlating approach under thermodynamic framework for better correlation purpose; under such circumstance, the solid–liquid equilibrium criteria approach may be useful in correlating the solubility [5,6,7]. In the present work we aimed at the development of new solubility models for the anthraquinone derivative which will be useful for supercritical dyeing process. In this work, we proposed two new models based on solid–liquid equilibrium criteria. Further, important exiting solubility models are also evaluated for all the compounds for the comparison purpose. The following section deals with existing solubility models considered in this study.

## 2. Existing Solubility Models

### 2.1. Empirical Models

#### 2.1.1. Chrastil Model

Chrastil et al. [8] proposed a semi empirical model based on solvate complex theory and have related the solubility of solute to density of supercritical fluid as follows:(1)$$S_{2}=\rho_{{\mathrm{ScCO}}_{2}}^{k}\exp\left(\frac{A_{1}}{T}+A_{2}\right)$$
where $S_{2}$ is the solute solubility in kg·m^−3^, *k* is the association number, $d_{1}$ is constant, and $d_{2}$ presents the function of enthalpy of solvation and vaporization. Equation (1) can be rewritten [9] to be mole fraction terms as follows:(2)$$y_{2}=\frac{{\left(\rho_{{\mathrm{ScCO}}_{2}}\right)}^{k-1}\;\exp\left(\frac{A_{1}}{T}+d_{2}\right)}{\left[1+{\left(\rho_{{\mathrm{ScCO}}_{2}}\right)}^{k-1}\;\exp\left(\frac{A_{1}}{T}+A_{2}\right)\right]}$$

#### 2.1.2. Adachi and Lu Model

Adachi and Lu (1983) [10] modified Chrastil’s equation by considering the quantity *k* to be density-dependent and the model can be written as:(3)$$y_{2}=\rho_{{\mathrm{ScCO}}_{2}}^{B_{1}+B_{2}\rho_{{\mathrm{ScCO}}_{2}}+B_{3}\rho_{{\mathrm{ScCO}}_{2}}{}^{2}}\exp\left(\frac{B_{4}}{T}+B_{5}\right)$$
where $y_{2}$ is the solute solubility in the mole fraction, the $B_{1}$ to $B_{5}$ are parameters constant.

#### 2.1.3. Mitra–Wilson Model

Mitra and Wilson (1991) [11] developed an empirical model for solubility of solute as a function of temperature and pressure:(4)$$\ln\;S_{2}=C_{1\;}\ln P+C_{2}T+C_{3}PT+C_{4}\frac{P}{T}+C_{5\;}$$

Here, *P* is the pressure system used in atm, and $C_{1}$ to $C_{5\;}$ are the constant parameters.

#### 2.1.4. Keshmiri Model

Keshmiri et al. (2014) [12] proposed the possible linear relationship between ln $y_{2}$ and $\ln\rho_{{\mathrm{ScCO}}_{2}}$ as the following expression:(5)$$\ln\;y_{2}=D_{1}+\frac{D_{2}}{T}+D_{3}P^{2}+(D_{4}+\frac{D_{5}}{T})\;\ln\rho_{{\mathrm{ScCO}}_{2}}$$
where *T* and *P* are the temperature and pressure system used, respectively. The $D_{1}$ to $D_{5}$ are constant parameters.

#### 2.1.5. Khansary Model

Subsequently, Khansary et al. (2015) [13] also developed a model relationship between ln $y_{2}$ and $\ln\rho_{{\mathrm{ScCO}}_{2}}$ as:(6)$$\ln\;y_{2}=\frac{E_{1}}{T}+E_{2}P+E_{3}\frac{P^{2}}{T}+(E_{4}+E_{5}P)\;\ln\rho_{{\mathrm{ScCO}}_{2}}$$

The $E_{1}$ to $E_{5}$ are constant parameters.

#### 2.1.6. Bian Model

Bian et al. (2016) [14] found a model with five constant parameters with relationship between solubility of solute ($y_{2})\;\mathrm{in}\;\mathrm{mole}\;\mathrm{fraction}$ and density, $\rho_{{\mathrm{ScCO}}_{2}}$, and obtained the following model:(7)$$\ln\;y_{2}=F_{1}+\frac{F_{2}}{T}+\frac{F_{3}\rho_{{\mathrm{ScCO}}_{2}}}{T}+(F_{4}+F_{5}\rho_{{\mathrm{ScCO}}_{2}})\ln\rho_{{\mathrm{ScCO}}_{2}}$$
where $F_{1}$ to $F_{5}$ are the model parameters.

#### 2.1.7. Garlapati and Madras Model

Garlapati and Madras [2] proposed an empirical model and related solute solubility to density of supercritical fluid as:(8)$$\ln\;y_{2}=G_{1}+\left(G_{2}+G_{3}\rho_{{\mathrm{ScCO}}_{2}}\right)\;\ln\rho_{{\mathrm{ScCO}}_{2}}+\frac{G_{4}}{T}+G_{5}\ln\rho_{{\mathrm{ScCO}}_{2}}T$$
where $G_{1}$ to $G_{5}$ are constant parameters.

#### 2.1.8. Reddy Model

Reddy et al. (2018) [15] proposed an empirical model based on degrees of freedom analysis as:(9)$$\;y_{2}=\left(H_{1}+H_{2}P_{r}\right)T_{r}^{2}+\left(H_{3}\;+H_{4}\;P_{r}\right)T_{r}+H_{5}$$
where $\;P_{r}$ and $T_{r}$ are reduced pressure of carbon dioxide ($\;P_{r}=\frac{P}{P_{c}})$ and reduced temperature of carbon dioxide ($\;T_{r}=\frac{T}{T_{c}})$, respectively. The $H_{1}$- $H_{5}$ are model constants. The $P_{c}$ and $T_{c}$ are critical pressure ($P_{c}$ = 7.387 MPa) and critical temperature ($T_{c}$ = 304.12 K), respectively.

### 2.2. Solid–Liquid Equilibrium Criteria Model

The behavior of solid solute in the liquid phase is determined by a ratio of the fugacity between pure liquid solute and the solid state at pressure (*P*) and temperature (*T*), which have reported elsewhere [4,16,17,18]. Moreover, the activity of substance obtained from the melting temperature and the melting enthalpy of compound. The activity coefficient of the substance can be represented by the regular solution model together with theory of Flory Huggins [6,17,19]. The solubility representation of the solute in ScCO_2_ is expressed by
(10)$$\ln\;y_{2}=\frac{\Delta H_{2}^{m}}{RT}\left(\frac{T}{T_{m}}-1\right)-\;\frac{v_{2}}{RT}{\left(\delta_{1}-\delta_{2}\right)}^{2}-\;\ln\left(\frac{v_{2}}{v_{1}}\right)-1+\frac{v_{2}}{v_{1}}$$
where $\Delta H_{2}^{m}$, $T_{m},$ and $v_{2}$ are the enthalpy of melting, melting temperature, and molar volume of the solute, respectively. These data are presented in Table 1. $v_{1}$ is the molar volume of ScCO_2_. $\Delta H_{2}^{m}$ and $v_{2}$ are estimated by Jain et al. method [20] and by Fedors method [21], respectively. The solubility parameter of ScCO_2_ ($\delta_{1})$ is calculated by Giddings method [22],
(11)$$\delta_{1}=\;8.032{\left(P_{c}/\mathrm{MPa}\right)}^{0.5}\left(\frac{\rho_{r}}{2.66}\right)$$
where $P_{c}$ is critical pressure ($P_{c}$ = 7.387 MPa), $\rho_{r}$ is the reduced density of CO_2_, it can be calculated by ρr=ρρc; the density of ScCO_2_, ρ, is obtained from website of NIST Web Book [23]. By assumption that the ScCO_2_ density depends on the solubility parameter of the solid solute ($\delta_{2}$), the correlation can be expressed as:(12)$$\delta_{2}=\;a+b\rho_{ScCO_{2}}^{c}$$
where $\rho_{{\mathrm{ScCO}}_{2}}$ is the density of ScCO_2_ in (mol/m^3^) obtained from website of NIST Web Book [23], *a*, *b,* and *c* are adjustable parameters.

## 3. New Models

### 3.1. Model 1

In solid–liquid equilibrium criteria, the supercritical phase is generally assumed as an expanded liquid consisting of infinite dissolved solute. At equilibrium, the solubility is expressed as [2,13,14]
(13)$$y_{2}=\frac{1}{\gamma_{2}^{\infty }}\frac{f_{2}^{S}}{f_{2}^{L}}$$

In Equation (13), $\gamma_{2}^{\infty }$ is solute activity coefficient at infinite dilution in supercritical fluid and $f_{2}^{S}$, $f_{2}^{L}$ are fugacity of solute in solid phase and supercritical fluid phase, respectively. From thermodynamics, pure solid to pure liquid fugacity ratio is expressed [24] as
(14)$$\frac{f_{2}^{S}}{f_{2}^{L}}=\exp\left[\frac{\Delta H_{2}^{m}}{RT}\left(\frac{T}{T_{m}}-1\right)-\int_{T_{m}}^{T}\frac{1}{RT^{2}}\left[\int_{T_{m}}^{T}\left[\Delta C_{p}\right]dT\right]dT\right]$$

In Equation (14), $\Delta C_{p}$ is known as difference in heat capacity between that of solid state minus liquid state, *R* is well known as universal gas constant. Combining Equations (13) and (14) gives Equation (15) for solubility for a special case where $\Delta C_{p}$ is constant.
(15)$$y_{2}=\frac{1}{\gamma_{2}^{\infty }}\exp\left[\frac{\Delta H_{2}^{m}}{RT}\left(\frac{T}{T_{m}}-1\right)-\frac{\Delta C_{p}}{R}\left[\ln\left(\frac{T}{T_{m}}\right)-T_{m}\left(\frac{1}{T_{m}}-\frac{1}{T}\right)\right]\right]$$

In Equation (15), the quantities $\Delta H_{2}^{m}$ and $T_{m}$ are constants for a given substance, therefore the exponential term in Equation (15) is written only in terms of temperature as
(16)$$y_{2}=\frac{1}{\gamma_{2}^{\infty }}\exp\left[a+\frac{b}{T}+c\ln(T)\right]$$

In Equation (16), $N_{1}=\frac{\Delta H_{2}^{m}}{RT_{m}}+\frac{\Delta C_{p}}{R}\left[\ln\left(T_{m}\right)+1\right]$,$N_{2}=\frac{-\Delta H_{2}^{m}}{R}-\frac{\Delta C_{p}T_{m}}{R}$ and $N_{3}=\frac{-\Delta C_{p}}{R}$.

The required activity coefficients in Equation (16) can be obtained from van Laar equation [24] as
(17)$$\ln\left(\gamma_{2}^{\infty }\right)=A_{21}{\left[\frac{A_{12}y_{1}}{A_{12}y_{1}+A_{21}y_{2}}\right]}^{2}$$

Equation (17) combined with Equation (16) would give the new model as
(18)$$y_{2}=\exp\left[N_{1}+\frac{N_{2}}{T}+N_{3}\ln(T)\right]/\exp\left(A_{21}{\left[\frac{A_{12}y_{1}}{A_{12}y_{1}+A_{21}y_{2}}\right]}^{2}\right)$$

Equation (18) represents the five parameter model derived based on solid and liquid phase equilibrium criterion and van Laar model for activity coefficient. In Equation (18), $N_{1}$, $N_{2}$, $N_{3}$, $A_{12}$ and $A_{21}$ are constants.

### 3.2. Model 2

In this model, the solid–liquid equilibrium criteria are the same as that of model 1. In place of pure solid to pure liquid fugacity ratio, a second order polynomial in temperature is considered [25]. The consideration may be justified from the actual expression for the fugacity ratio [26,27], which is
(19)$$\ln\left(\frac{f_{s}^{}}{f_{L}}\right)=\frac{\Delta H_{2}^{m}}{R}\left(\frac{1}{T}-\frac{1}{T_{m}}\right)-\frac{1}{RT}\int_{T_{m}}^{T}\Delta C_{p}dT+\frac{1}{R}\int_{T_{m}}^{T}\frac{\Delta C_{p}}{T}dT+\int_{P_{2}^{sat}}^{P}\frac{v_{2}}{RT}dP$$

Equation (19) gross form is a polynomial in temperature. The polynomial term (for temperature dependence) in literature is also observed with the work presented by Nordström and Rasmuson [25], who fitted the solubility of salicylamide in various solvents at normal pressures. Therefore, fugacity ratio in this work is expressed as a second order polynomial in terms of temperature as exp (*A* + *B*/*T* + *C*/*T*^2^). Therefore, the final expression for solubility is
(20)$$y_{2}=\frac{1}{\gamma_{2}^{\infty }}\exp\left[A+\frac{B}{T}+\frac{C}{T^{2}}\right]$$

The required activity coefficients in Equation (20) can be obtained from van Laar equation as in Equation (17). Equation (17) combined with Equation (20) would give the new model as
(21)$$y_{2}=\exp\left[A+\frac{B}{T}+\frac{C}{T^{2}}\right]/\exp\left(A_{21}{\left[\frac{A_{12}y_{1}}{A_{12}y_{1}+A_{21}y_{2}}\right]}^{2}\right)$$

Equation (21) represents the five parameter model derived based on solid and liquid phase equilibrium criterion and van Laar model for activity coefficient. In Equation (21) *A*, *B*, *C*, *A*_12_, and *A*_21_ are constants.

## 4. Methodology

We used fminsearch algorithm which uses the Nelder–Mead simplex as described by Lagarias et al. [28] built in MATLAB software (R2019b) student version to fit models and experimental data collected from literature. Furthermore, we also inspected the quality of modeling through various entities such as correlation coefficient (*R*^2^), adjusted *R*^2^ (Adj. *R*^2^), root mean square deviation (*RMSE*), sum of squares due to error (*SSE*), and the overall average absolute relative deviation (*AARD*) between experimental data and calculated results. The *R^2^*, Adj. *R^2^*, *SSE*, and *RMSE* are evaluated using the following formulas [29]
(22)$$AARD/\left(\%\right)=\frac{100}{N_{i}}\overset{N_{i}}{\underset{i=1}{\sum }}\frac{\left|y_{2}^{\mathrm{cal}}-y_{2}^{\exp}\right|}{y_{2}^{\exp}}$$
(23)$$R^{2}=1-\frac{\sum_{i=1}^{N_{i}}{\left(y_{2}^{\exp}-y_{2}^{\mathrm{cal}}\right)}^{2}}{\sum_{i=1}^{N_{i}}{\left(y_{2}^{\bar{\exp}}-y_{2}^{\mathrm{cal}}\right)}^{2}}$$
(24)$$\mathrm{Adj}.\;R^{2}=R^{2}-\frac{Q\left(1-R^{2}\right)}{N_{i}-Q-1}$$
(25)$$SSE=\left[\overset{N_{i}}{\underset{i=1}{\sum }}{\left(y_{2}^{\exp}-y_{2}^{\mathrm{cal}}\right)}^{2}\right]$$
(26)RMSE=1Ni∑i=1Niy2exp−y2cal212

In Equation (22), $y_{2}^{\mathrm{cal}}$ and $y_{2}^{\exp}$ represent the mole fraction of calculated and experimental solubility’s values, respectively. $y_{2}^{\bar{\exp}}$ is the global mean value of experimental data in mole fraction.

Statistical comparison of models is essential to ensure the success of the new model. In order to this achieve this, the well-known Akaike’s Information Criterion (AIC) proposed by Akaike [30,31] has been used. AIC is expressed as
(27)$$\mathrm{AIC}=N\;\ln\left(\frac{SSE}{N}\right)+2K$$

In Equation (27), *K* is number of parameter constants, *N* is number of data points, *SSE* is the sum of squares due to error. Importantly, AIC is number of adjustable parameters of the individual model.

## 5. Results and Discussion

In this study, we propose two new solid–liquid equilibrium criteria models to correlate solubility of solid in supercritical carbon dioxide. The accuracy of the proposed models is evaluated by correlating 25 anthraquinone derivative compounds available in the literature. The correlating ability of the new models are evaluated in terms of: *AARD*, *R^2^*, Adj. *R^2^*, *SSE*, and *RMSE*. There are more than 25 models available in literature [29] for correlating solubility of solids in supercritical fluids. However, for comparison purposes, we have considered Chrastil model, Adachi and Lu model, Mitra—Wilson model, Keshmiri et al. model, Khansary et al. model, Garlapati and Madras model, Reddy et al., model, and one existing three parameters solid–liquid equilibrium model. These are grouped as three parameter models and five parameter models. Table 2 shows the information of the 25 anthraquinone derivatives considered in this study. Table 2 shows the solubility range and references [4,16,17,18,32,33,34,35,36,37,38] from which the data are obtained. Table 1 shows the physical properties such as melting point, melting enthalpy, and molar volume of the solutes. For some compounds, these properties are not available and for such compounds we have used the Jain et al. method [20] and Fedors method [21] for evaluating the melting enthalpy and solute molar volumes, respectively. The constant parameters of literature models considered, Chrastil, Adachi-Lu, Mitra—Wilson, Keshmiri et al., Khansary et al., Bian et al., Garlapati—Madras, and Reddy et al., are listed in the Appendix A. Table 3 shows the correlation results of the three parameter solid–liquid equilibrium model. Table 4 shows the correlation constants of the new model 1. Table 5 shows the correlation constants of the new model 2. Table 6 shows the overall mean statistical parameters of various solubility models. From Table 6, it is clear that the proposed models show the lowest AARD. The new model 1 shows an overall AARD% of 6.538 and the second model (new model 2) shows an overall AARD% of 6.377. The two models proposed in this work are observed to perform the correlation on a par. Although they look different in functional form, their correlation ability is matching well. This correlating matching ability may be attributed to its oneness in their functional form.

To know the efficacy of the proposed models, further analysis is carried out with paired t-test and Akaike’s Information Criterion (AIC). Table 7 shows the paired t-test (paired t-test, *p* < 0.05) results for AARD, *R*^2^, and Adj. *R*^2^. From the results, it is clear that AARDs of the new models are statistically significant. Table 7 shows the paired t-test results for SSE and *RMSE*. From the results, it is clear that SSEs of the new models are not statistically significant. *R*^2^, Adj.*R*^2^, and *RMSE* are showing mixed results and hence we could not infer any statistical meaning from them such as significant or not significant. Table 8 shows AIC information of the proposed models and literature models. From Table 4 and Table 5, the new models are significantly different at 95% confidence level (paired t-test, *p* < 0.05). Table 8 of AIC information shows that among all models, the new models are having lower AIC values. The AIC value for the new model 1 is −730.59, and for the new model 2 is −1177.56. The lower AIC value indicates the goodness of the new models and we conclude that those models are superior to other models considered in the work.

From new model 1 constants, one can calculate melting temperature and melting enthalpies. The back calculations are a bit tricky and we need to use a root finding method to calculate melting temperature and then melting enthalpy is estimated. The calculated values are reported in Table 9. It is observed that the melting temperature is much lower than actual values (Table 1); whereas the melting enthalpies for few compounds (Compound numbers 9, 11, 20, and 24) are magnitude wise matching with the computed values reported in Table 1 (Jain et al. method). This disparity may be attributed to use of approximate empirical expression for the fugacity ratio for the development of the solubility expression. Probably, exact expression would give better results and this is out of the scope of the present work.

To illustrate the ability of the proposed models, solubility data of 1-amino-2,3-dimethyl-9,10-anthraquinone in supercritical carbon dioxide were selected as illustrated in Figure 1, Figure 2, Figure 3 and Figure 4, respectively. In another illustration, we selected Red 15 (1-amino-4-hydroxyanthraquinone) to show the goodness of the new model 1 and the new model 2 (Figure 5 and Figure 6). In Figure 7, the global mean AARD% values of all models is depicted. In terms of global mean AARD, the overall order for the ability correlating of the models is: new model 2 > new model 1 > Adachi and Lu > Garlapati and Madras > Keshmiri et al. > Chrastil > Khansary et al. > Bian et al. > Mitra and Wilson > SLE model > Reddy et al.

## 6. Conclusions

The new models developed in this study may be useful in correlating solubility data of any compound in supercritical fluid. A comparison between the proposed models and some specific literature models (particularly three and five parameters constants) was made to correlate the solubility of 25 anthraquinone derivatives. The results showed that the proposed models exhibited excellent agreement with those experimental data in the literature and that the proposed models are superior to all of the other models considered in the present work with AARD of 6.538% for new model 1, and AARD of 6.377% for new model 2. The new models of this work can be used for modeling solubility of any other system.

## Figures and Tables

**Figure 1 molecules-26-00460-f001:**
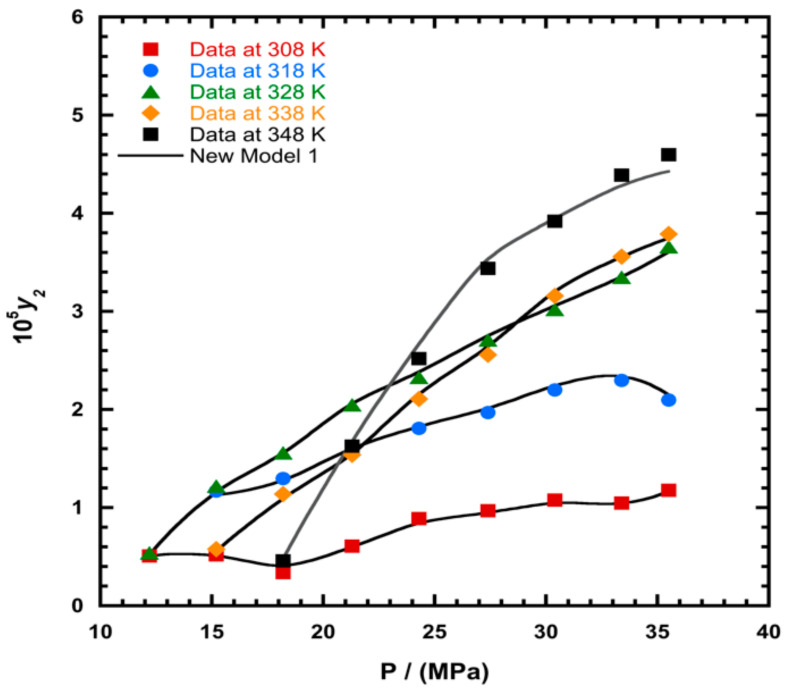
Plot of mole fraction (*y_2_*) as a function of pressure (P/MPa) for 1-amino-2,3-dimethyl-9,10-anthraquinone, the solid line represents the proposed model 1 (Equation (18)).

**Figure 2 molecules-26-00460-f002:**
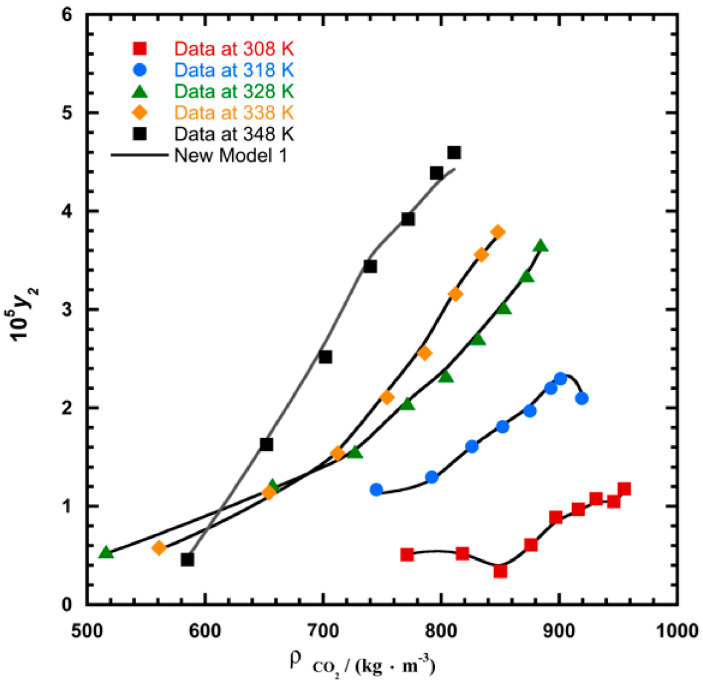
Plot of mole fraction (*y_2_*) as a function of density ρ/(kg.m^−3^) for 1-amino-2,3-dimethyl-9,10-anthraquinone, the solid line represents the proposed model 1 (Equation (18)).

**Figure 3 molecules-26-00460-f003:**
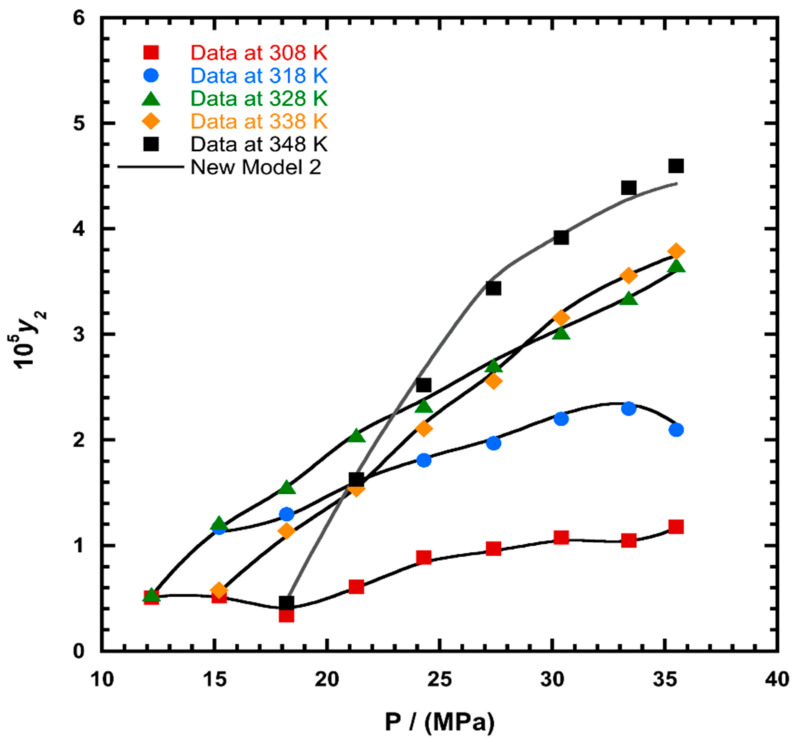
Plot of mole fraction (*y_2_*) as a function of pressure (P/MPa) for 1-amino-2,3-dimethyl-9,10-anthraquinone, the solid line represents the proposed model 2 (Equation (21)).

**Figure 4 molecules-26-00460-f004:**
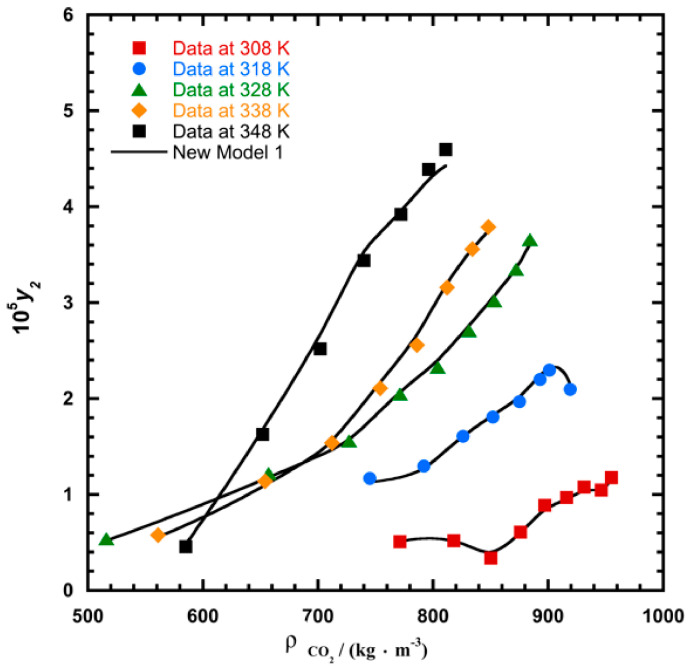
Plot of mole fraction (*y_2_*) as a function of density ρ/(kg.m^−3^) for 1-amino-2,3-dimethyl-9,10-anthraquinone, the solid line represents the proposed model 2 (Equation (21)).

**Figure 5 molecules-26-00460-f005:**
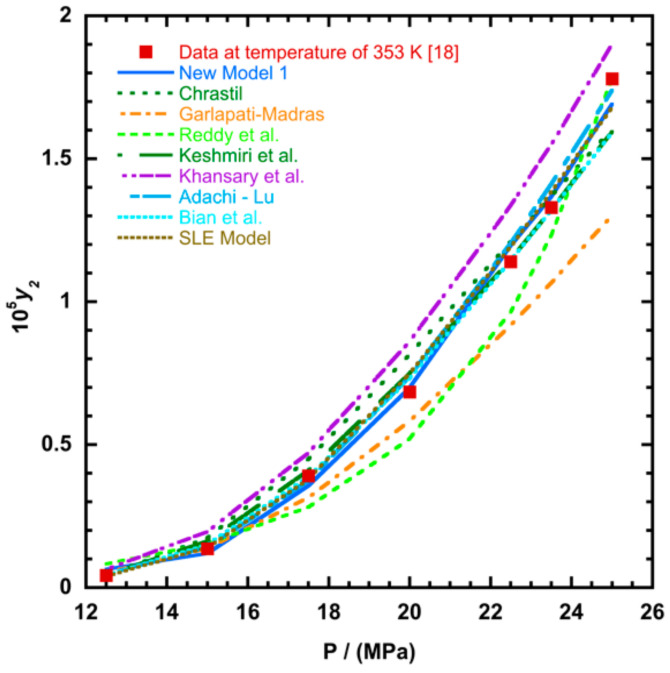
Plot of mole fraction solubility of Red 15 (1-amino-4-hydroxyanthraquinone) against pressure (P/MPa), the solid line presents the new model 1 (Equation (18)).

**Figure 6 molecules-26-00460-f006:**
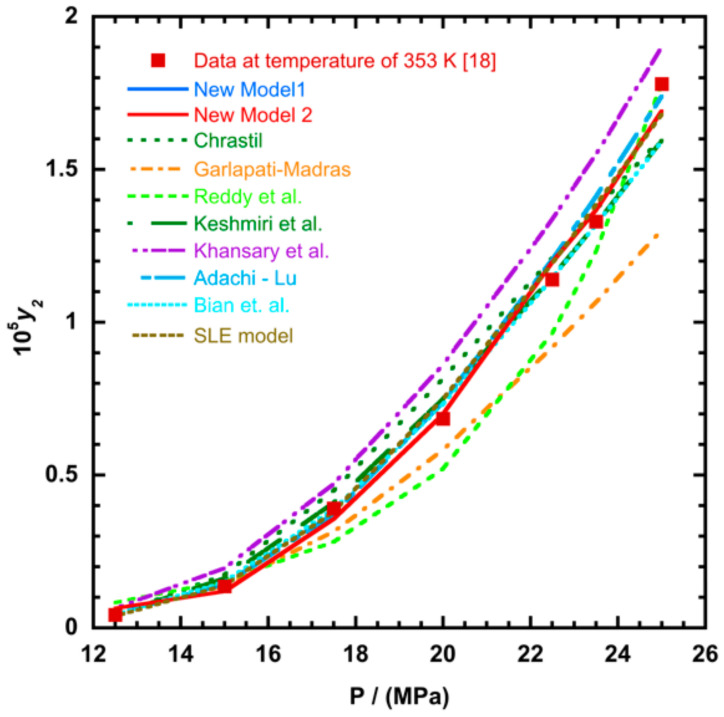
Plot of mole fraction solubility of Red 15 (1-amino-4-hydroxyanthraquinone) against pressure (P/MPa), the solid lines present the new model 1 (Equation (18)) and the new model 2 (Equation (21)), respectively.

**Figure 7 molecules-26-00460-f007:**
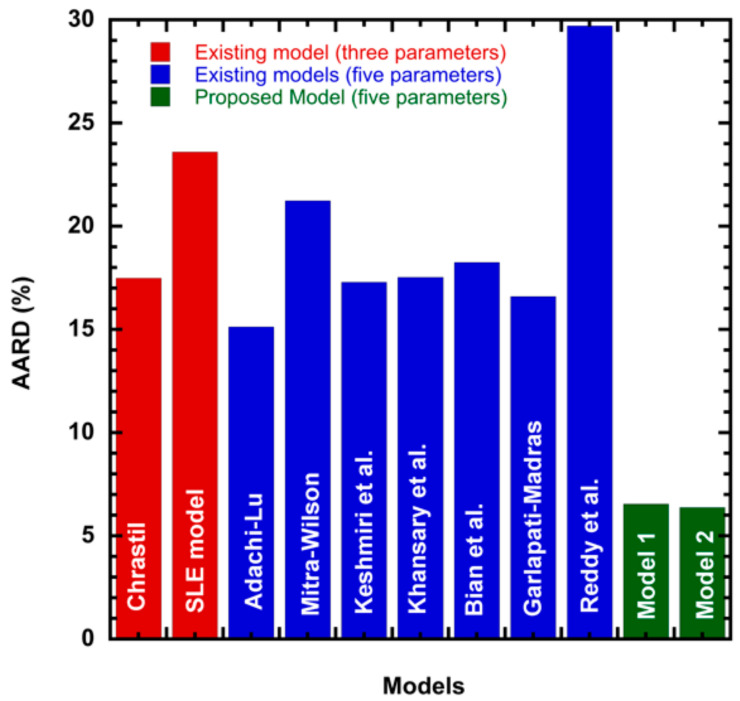
Global mean AARD% of literature models and the proposed model.

**Table 1 molecules-26-00460-t001:** Physical properties of the compounds.

Serial Number & Name		*T* _m_	$\Delta H_{2}^{m}$^b^ (KJ/mol)	*v*_2_·10^4 c^ (m^3^/mol)
1.	C.I. disperse blue 3	453.6 ^a^	35.48	2.172
2.	Blue 1	599.74 ^b^	34.42	1.894
3.	1,4-dihydroxy-9,10-anthraquinone	469.15 ^a^	27.81	1.665
4.	1-Hydroxy-4-(prop-2-enyloxy)-9,10-anthraquinone	463.38 ^b^	31.74	2.030
5.	1,4-bis(prop-2′-enyloxy)-9,10-anthraquinone	448.17 ^b^	35.67	2.280
6.	1-amino-2-methylanthraquinone	478.15 ^a^	24.12	1.789
7.	1- amino-2-ethyl-9,10-anthraquinone	427.15 ^a^	26.95	1.732
8.	1-amino-2,3-dimethylanthraquinone	486.15 ^a^	24.60	1.790
9.	1-hydroxy-9,10-anthraquinone	599.28 ^a^	23.92	1.610
10.	1-hydroxy-2-methylanthraquinone	458.15 ^a^	24.40	1.759
11.	1-hydroxy-2-(methoxy methyl)anthraquinone	433.94 ^b^	29.72	1.964
12.	1-hydroxyl-2-(ethoxy methyl)anthraquinone	401.15 ^a^	32.55	2.113
13.	1-hydroxy-2-(1-propoxy methyl)anthraquinone	424.0 ^b^	35.39	2.263
14.	1-hydroxy-2-(1-butoxymethyl) anthraquinone	389.74 ^b^	38.22	2.412
15.	1-hydroxy-2-(n-amyloxy methyl) anthraquinone	418.64 ^b^	41.06	2.561
16.	Quinizarin	469.15^a^	27.81	1.665
17.	Violet 1(1,4-diaminoanthraquinone)	539.15 ^a^	27.23	1.178
18.	Blue 59 (1,4-bis (ethyl amino)anthraquinone)	471.15 ^a^	33.04	1.880
19.	Red 15 (1-amino-4-hydroxyanthraquinone)	489.15 ^a^	27.51	1.116
20.	1 hydroxy-4-nitroanthraquinone	540 ^a^	26.61	1.214
21	1,8-dihidroxy-4,5-dinitroanthraquinone	573.1 ^a^	33.19	1.254
22.	1,4 diamino-2,3-dichloroanthraquinone	576 ^a^	29.38	1.758
23.	1-aminoanthraquinone	526 ^a^	23.63	1.176
24.	1-nitroanthraquinone	505.5 ^a^	22.73	1.554
25.	C.I. Disperse orange 11	478.15 ^a^	24.12	1.789

**^a^** From CAS databased. (https://scifinder.cas.org/scifinder/view/scifinder/scifinderExplore.jsf). ^b^ Estimated by Jain et al. method [20]. ^c^ Estimated by Fedors method [21].

**Table 2 molecules-26-00460-t002:** Solubility information of the compounds.

Serial Number and Name		Chemical Structure	Solubility Range $y_{2}$ × 10^6^	T(K) and P(MPa) Range	N	Reference
1.	C.I. disperse blue 3	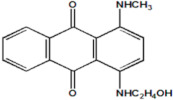	0.68–63.575	(323.7–413.7); (10.51–32.98)	23	[33]
2.	Blue 1	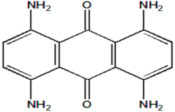	6.63–44.5	(333.3–373.2); (20–40)	18	[34]
3.	1,4-dihydroxy-9,10-anthraquinone	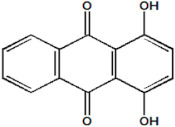	13–314	(308–348); (12.16–40.53)	40	[35,36]
4.	1-Hydroxy-4-(prop-2′-enyloxy)-9,10-anthraquinone	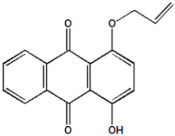	9–498	(308–348); (12.16–40.53)	38	[36]
5.	1,4-bis(prop-2′-enyloxy)-9,10-anthraquinone	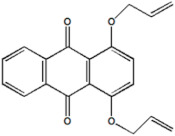	2–200	(308–348); (12.16–40.53)	34	[36]
6.	1-amino-2-methylanthraquinone	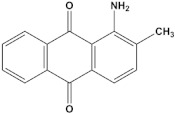	4.6–109.6	(308–348); (12.2–35.5)	43	[37]
7.	1- amino-2-ethyl-9,10-anthraquinone	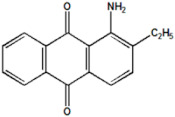	2.6–77.8	(308–348); (12.2–35.5)	43	[37]
8.	1-amino-2,3-dimethylanthraquinone	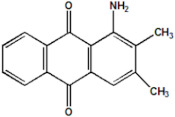	4.6–37.9	(308–348); (12.2–35.5)	41	[37]
9.	1-hydroxy-9,10-anthraquinone	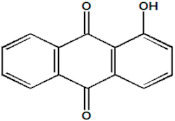	30–445	(308–348); (12.2–35.5)	45	[38]
10.	1-hydroxy-2-methylanthraquinone	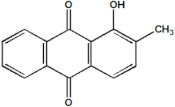	9–737	(308–348); (12.2–35.5)	45	[38]
11.	1-hydroxy-2-(methoxy methyl)anthraquinone	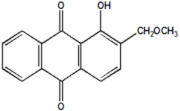	1–537	(308–348); (12.2–35.5)	45	[38]
12.	1-hydroxyl-2-(ethoxy methyl)anthraquinone	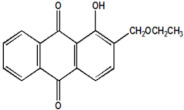	23–1100	(308–348); (12.2–35.5)	45	[38]
13.	1-hydroxy-2-(1-propoxy methyl)anthraquinone	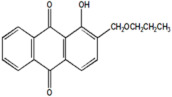	103–1676	(308–348); (12.2–35.5)	45	[38]
14.	1-hydroxy-2-(1-butoxymethyl) anthraquinone	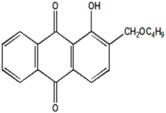	82–2699	(308–348); (12.2–35.5)	45	[38]
15.	1-hydroxy-2-(n-amyloxy methyl) anthraquinone	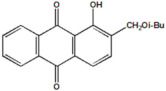	38–2640	(308–348); (12.2–35.5)	45	[38]
16.	Quinizarin	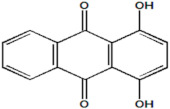	69–6940	(353.2–393.2); (12–30)	15	[35,36]
17.	Violet 1(1,4-diaminoanthraquinone)	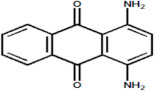	0.13–2.61	(323.15–383.15); (15–25)	15	[16]
18.	Blue 59 (1,4-bis (ethyl amino)anthraquinone)	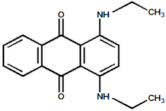	0.218–14.9	(323.15–383.15); (12.5–25)	26	[16]
19.	Red 15 (1-amino-4-hydroxyanthraquinone)	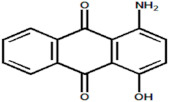	1.84–24.5	(323.15–383.15); (12.5–25)	20	[18]
20.	1 hydroxy-4-nitroanthraquinone	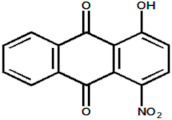	1.22–8.64	(323.15–383.15); (15–25)	15	[18]
21	1,8-dihidroxy-4,5-dinitroanthraquinone	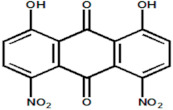	0.168–1.12	(323.15–383.15); (15–25)	15	[4]
22.	1,4 diamino-2,3-dichloroanthraquinone	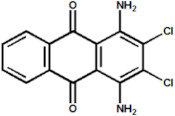	0.053–5.24	(323.15–383.15); (12.5–25)	18	[4]
23.	1-aminoanthraquinone	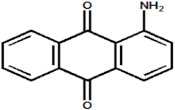	0.55–35.1	(323.15–383.15); (12.5–25)	18	[17]
24.	1-nitroanthraquinone	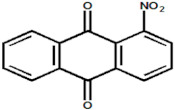	0.984–25.2	(323.15–383.15); (12.5–25)	18	[17]
25.	C.I. Disperse orange 11	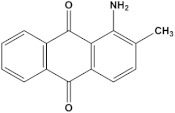	0.58–30.3	(323.15–383.15); (12–25)	12	[32]

**Table 3 molecules-26-00460-t003:** Correlation results of the three parameter solid–liquid equilibrium model (Equations (10)–(12)).

Sl.No*	*a*	*b*	*c*	AARD%
1	16,983	0.111170	1.15140	55.328
2	14,423	8.083700	0.76039	34.399
3	14,867	0.524320	1.01780	12.935
4	13,786	0.390040	1.04700	14.982
5	15,545	0.025248	1.30360	36.224
6	14,282	1.375900	0.92731	16.165
7	17,441	0.022117	1.31830	25.785
8	19,195	0.002538	1.52380	23.006
9	13,508	0.477910	1.03180	12.740
10	16,683	0.021064	1.32190	8.198
11	15,754	0.000044	1.97910	82.871
12	13,395	0.693360	0.99182	18.228
13	13,574	0.153700	1.13460	16.541
14	10,512	2.867000	0.86305	17.309
15	13,936	0.081591	1.18840	36.363
16	16,800	0.000017	2.01370	32.140
17	22,206	0.000122	1.59150	39.940
18	18,628	0.010357	1.38690	8.268
19	22,846	0.009591	1.38010	6.304
20	22,936	0.001514	1.56360	11.761
21	23,417	0.001591	1.55150	28.828
22	19,181	0.005504	1.45030	14.566
23	22,218	0.002795	1.50580	5.625
24	18,613	0.096199	1.17250	10.777
25	18,221	0.010624	1.39170	20.698

Sl.No*: Serial number and name same as Table 2.

**Table 4 molecules-26-00460-t004:** Correlation constants of the new model 1 (Equation (18)).

Sl.No*	*A* _12_	*A*_21_·10^5^	*N* _1_	*N* _2_	*N* _3_	AARD%	*R* ^2^	Adj.*R*^2^	*RMSE*·10^7^	*SSE*·10^16^
1	4.6317	7.4408	−123.220	5849.4	16.501	10.3260	0.925	0.904	30.93	2,295,600
2	3.6237	0.1369	−22.280	491.4	1.197	1.2124	0.998	0.997	0.04	2.85400
3	3.8856	56.018	−119.110	5355.7	16.390	4.3482	0.980	0.977	76.09	23,159,000
4	4.1249	66.839	−255.500	11861.0	36.544	7.8210	0.802	0.773	185.89	138,230,000
5	4.4232	34.492	352.360	−17338.0	−53.119	7.8357	0.905	0.891	96.46	37,222,000
6	3.9076	20.069	−35.941	1222.0	4.032	4.6621	0.949	0.943	28.25	3,592,400
7	4.3202	11.976	−44.086	1418.5	5.251	5.8040	0.941	9.340	29.90	4,023,100
8	3.7536	10.665	−133.510	5959.7	18.268	2.3299	0.981	0.979	6.39	183,520
9	3.7520	100.30	−65.421	2831.4	8.549	2.5184	0.960	0.954	66.34	19,807,000
10	4.1139	100.84	−118.880	5233.1	16.522	8.1326	0.945	0.938	308.08	427,110,000
11	3.6917	132.15	−81.270	3572.1	10.940	8.1276	0.843	0.823	309.64	431,440,000
12	3.8667	183.28	−54.582	2282.9	7.074	4.9246	0.960	0.955	300.20	405,550,000
13	3.8025	327.28	−73.225	3236.9	9.889	3.6837	0.963	0.958	333.17	499,500,000
14	4.0108	399.95	−203.750	9484.9	29.170	6.1019	0.964	0.960	825.19	3,064,200,000
15	4.2943	294.42	−80.585	3362.6	11.087	9.7765	0.919	0.909	1267.30	7,226,900,000
16	4.8875	1186.7	967.570	−52133.0	−140.540	11.4640	0.906	0.867	8137.90	119,200,000,000
17	3.8754	0.4939	−95.533	4234.2	12.097	4.6045	0.946	0.924	0.66	657.72
18	4.2476	2.0376	−114.860	5180.0	15.189	10.5440	0.967	0.960	6.82	139,560
19	4.1111	3.8448	−141.350	6609.0	19.113	8.9540	0.922	0.879	9.75	199,440
20	3.7799	1.8628	−78.114	3418.2	9.746	3.1992	0.993	0.989	1.63	3990.60
21	3.5828	0.3210	−31.261	922.3	2.662	1.1563	0.973	0.958	0.10	13.78
22	4.3364	0.5898	−148.990	6728.0	20.046	10.9930	0.891	0.831	2.82	14339
23	4.1502	4.8759	−128.930	5935.5	17.365	9.4322	0.903	0.848	14.82	395,570
24	3.9199	4.3919	−67.609	2866.9	8.373	5.3605	0.977	0.965	7.33	96,592
25	4.3723	3.6415	28.875	−2317.9	−5.576	10.1480	0.970	0.953	16.25	316,930

Sl.No*: Serial number and name same as Table 2.

**Table 5 molecules-26-00460-t005:** Correlation constants of the new model 2 (Equation (21)).

Sl.No*	*A* _12_	*A*_21_·10^5^	*A*	*B*	*C*·10^−5^	AARD/%	*R* ^2^	Adj.*R*^2^	*RMSE*·10^12^	*SSE*·10^6^
1	4.6319	7.461	−1.0778	−6194.3	10.941	10.347	0.925	0.904	229.99000	3.0956
2	3.6237	0.137	−13.461	−353.46	0.744	1.2124	1.000	1.000	0.00029	0.0040
3	3.8855	56.020	0.35304	−5347.8	8.728	4.3504	0.980	0.977	2319.70	7.6153
4	4.1251	66.837	10.906	−12028	19.499	7.8214	0.947	0.940	13855.00	18.6110
5	4.4208	34.554	−34.627	17225	−28.081	7.8227	0.905	0.891	3719.90	9.6436
6	3.9077	20.070	−6.642	−1350.7	2.048	4.6627	0.949	0.943	360.00	2.8284
7	4.3203	11.977	−5.9057	−1950	2.697	5.8043	0.941	0.934	403.16	2.9932
8	3.7542	10.660	−0.2361	−6047.9	9.856	2.3265	0.981	0.979	18.39	0.6393
9	3.752	100.310	−3.0674	−2776.4	4.593	2.5168	0.961	0.956	1983.30	6.6387
10	4.114	100.850	1.5165	−5534	8.762	8.1321	0.945	0.938	42,770.00	30.8290
11	3.7153	120.250	0.29924	−4765.9	7.677	4.0867	0.984	0.982	2753.40	7.8222
12	3.8668	183.240	−2.9085	−2409.6	3.886	4.9225	0.960	0.954	40,519	30.0070
13	3.8026	327.230	−1.0388	−3291.2	5.381	3.6796	0.963	0.958	49,885	33.2950
14	4.0108	400.240	8.7869	−9511.2	15.448	6.1073	0.965	0.960	307,640	82.6830
15	4.2944	294.450	−0.13121	−3641.3	5.518	9.7817	0.919	0.909	725,400	126.960
16	4.8875	1186.800	−75.478	52,674	−97.606	11.464	0.906	0.867	11,919,000	813.740
17	3.8754	0.494	−6.4227	−4295.6	7.500	4.6045	0.946	0.916	0.06577	0.0662
18	4.2476	2.038	−2.9703	−5529.4	9.416	10.544	0.967	0.960	13.9560	0.6820
19	4.1111	3.845	−0.55598	−6866.1	11.846	8.954	0.922	0.896	19.9440	0.9745
20	3.7799	1.863	−6.3245	−3452.8	6.041	3.1992	0.993	0.989	0.39906	0.1631
21	3.5828	0.321	−11.654	−954.12	1.649	1.1563	0.973	0.958	0.00138	0.0096
22	4.3364	0.590	−1.3261	−7404.6	12.425	10.993	0.891	0.846	1.43390	0.2823
23	4.1502	4.876	−1.0106	−6311.3	10.770	9.4324	0.903	0.862	39.5560	1.4824
24	3.9199	4.392	−5.9336	−3036	5.190	5.3605	0.977	0.968	9.65920	0.7326
25	4.3723	3.642	−12.201	1613.4	−3.456	10.148	0.970	0.945	31.6930	1.6251

Sl.No*: Serial number and name same as Table 2.

**Table 6 molecules-26-00460-t006:** Overall mean statistical parameters of solubility models.

Model	No. of Constants	*R* ^2^	Adj. *R*^2^	*SSE*	*RMSE*	AARD %
Chrastil	3	0.89690	0.89295	1.30 ×10^−6^	1.01 ×10^−4^	17.485
Adachi-Lu	5	0.89850	0.89780	1.27	7.69 ×10^−2^	15.130
Mitra-Wilson	5	0.87990	0.87560	3.539	1.70 ×10^−1^	21.240
Keshmiri et al.	5	0.89100	0.35800	2.75 ×10^−7^	1.00 ×10^−4^	17.298
Khansary et al.	5	0.89100	0.88700	6.67 ×10^−7^	6.24 ×10^−5^	17.530
Bian et al.	5	0.89644	0.89278	2.69 ×10^−7^	4.97 ×10^−5^	18.251
Garlapati-Madras	5	0.87800	0.87300	2.01 ×10^−7^	4.53 ×10^−5^	16.599
Reddy et al.	5	0.75600	0.74400	3.92 ×10^−6^	1.50 ×10^−4^	29.711
SLE model	3	0.88985	0.86064	4.44 ×10^−6^	1.77 ×10^−4^	23.599
New Model 1	5	0.93930	0.92270	5.26 ×10^−7^	4.825 ×10^−5^	6.538
New Model 2	5	0.95092	0.93728	4.73 ×10^−4^	5.244 ×10^−7^	6.377

**Table 7 molecules-26-00460-t007:** a. Paired *t*-test results for averaged absolute relative deviation (AARD), *R*^2^ and Adj.*R*^2^, b. Paired *t*-test results for sum of squares due to error (SSE) and *RMSE.*

**Paired *t*-test Results for AARD, *R*^2^ and Adj.*R*^2^**						
**Models**	**AARD**		***R*^2^**		**Adj. *R*^2^**	
	**New Model 1**	**New Model 2**	**New Model 1**	**New Model 2**	**New Model 1**	**New Model 2**
Chrastil	S	S	NS	NS	NS	NS
Adachi-Lu	S	S	NS	NS	NS	NS
Mitra-Wilson	S	S	S	S	NS	NS
Keshmiri et al.	S	S	NS	NS	NS	NS
Khansary et al.	S	S	S	S	S	NS
Bian et al.	S	S	NS	NS	NS	NS
Garlapati-Madras	S	S	S	S	NS	NS
Reddy et al.	S	S	S	S	NS	S
SLE model	S	S	S	S	NS	NS
**Paired *t*-test Results for SSE and *RMSE***						
**Models**	**SSE**			***RMSE***		
	**New Model 1**		**New Model 2**	**New Model 1**		**New Model 2**
Chrastil	NS		NS	NS		NS
Adachi-Lu	NS		NS	S		NS
Mitra-Wilson	NS		NS	S		S
Keshmiri et al.	NS		NS	NS		NS
Khansary et al.	NS		NS	S		NS
Bian et al.	NS		NS	NS		NS
Garlapati-Madras	NS		NS	NS		NS
Reddy et al.	NS		NS	NS		NS
SLE model	NS		NS	NS		NS

NS: Not significant; S: Significant.

**Table 8 molecules-26-00460-t008:** AIC information of the proposed models and literature models.

Sl.No*	Equation (18)	Equation (21)	Equation (2)	Equation (3)	Equation (4)	Equation (5)	Equation (6)	Equation (7)	Equation (8)	Equation (9)	SLE (Equations (10)−(12))
1	−572.60	−883.48	−543.41	−547.95	−544.26	−529.96	−533.91	−548.37	−521.14	−520.16	−502.02916
2	−686.29	−804.56	−604.41	−296.42	−285.04	−603.40	−608.51	−587.68	−602.21	−583.75	−577.05251
3	−932.89	−1530.00	−968.16	−303.79	−230.96	−975.80	−948.67	−929.28	−880.29	−824.08	−904.08437
4	−815.91	−1417.10	−711.57	−53.04	−44.02	−700.99	−706.31	−693.52	−698.85	−672.92	−692.25585
5	−769.80	−1285.45	−730.70	−140.86	−99.47	−731.25	−724.22	−704.18	−716.47	−677.58	−656.16322
6	−1086.85	−1691.20	−1051.84	−326.98	−284.62	−1041.34	−1035.41	−1010.52	−1017.78	−951.70	−999.79284
7	−1081.98	−1688.77	−1001.05	−257.98	−232.93	−1006.04	−1001.16	−1009.39	−999.11	−945.54	−960.43326
8	−1155.83	−1671.09	−1022.17	−331.42	−313.68	−1020.25	−1035.60	−1005.96	−977.98	−980.62	−1008.7995
9	−1063.09	−1733.98	−973.50	−220.08	−210.03	−954.59	−955.82	−973.06	−975.24	−894.61	−974.00663
10	−924.90	−1664.88	−970.35	−215.20	−156.93	−961.05	−951.72	−929.51	−928.08	−832.69	−975.57241
11	−924.44	−1726.60	−815.67	−760.86	−930.34	−820.67	−943.42	−811.84	−808.04	−986.59	−921.64403
12	−927.23	−1666.10	−913.68	−153.10	−143.13	−916.41	−889.74	−923.98	−887.23	−849.04	−900.93092
13	−917.85	−1661.42	−876.78	−128.34	−80.13	−883.27	−875.88	−866.40	−821.62	−790.88	−871.09925
14	−836.22	−1620.49	−815.23	−26.09	−22.74	−795.27	−789.52	−795.24	−845.10	−743.41	−815.42235
15	−797.61	−1601.19	−790.56	−791.01	−764.98	−787.64	−774.85	−789.17	−793.49	−732.22	−773.78737
16	−200.68	−482.72	−218.20	−189.25	25.92	−216.10	−196.21	−217.34	−223.00	−170.32	−202.70016
17	−485.91	−623.97	−270.41	−234.37	−238.65	−486.03	−469.66	−494.58	−491.75	−463.79	−173.59318
18	−724.58	−1042.54	−758.18	−324.42	−304.19	−729.89	−732.67	−755.40	−699.79	−705.83	−763.27385
19	−542.68	−787.26	−265.77	−241.05	−228.48	−547.15	−532.15	−545.48	−514.00	−533.35	−569.89688
20	−458.87	−610.44	−438.25	−182.81	−199.53	−429.90	−419.25	−456.83	−432.72	−433.09	−441.6821
21	−543.89	−652.93	−485.64	−245.06	−236.68	−470.10	−480.91	−486.40	−479.42	−481.78	−467.63919
22	−532.90	−727.94	−559.09	−248.76	−247.35	−538.41	−527.98	−556.60	−539.48	−528.07	−570.46594
23	−473.19	−698.09	−472.09	−182.95	−178.74	−459.60	−456.90	−486.09	−446.49	−435.79	−486.12245
24	−498.56	−710.77	−485.77	−195.10	−185.25	−475.73	−471.13	−497.48	−476.43	−481.84	−505.23963
25	−309.92	−456.09	−303.97	−110.80	−97.41	−298.57	−295.69	−298.64	−275.30	−287.64	−295.23155
Overall	−730.59	−1177.56	−681.86	−268.31	−249.34	−695.18	−694.29	−694.92	−682.04	−660.29	−680.36

Sl.No*: Serial number and name same as Table 2.

**Table 9 molecules-26-00460-t009:** Computed *T_m_* and $\Delta H_{2}^{m}$ from new model 1.

Sl.No*	*N* _1_	*N* _2_	*N* _3_	Tm (K) ^a^	$-\Delta H_{2}^{m}(J/\mathrm{mol}){\;}^{b}$
1	−123.22	5849.4	16.501	140.5761	46,711.00
2	−22.28	491.4	1.197	26.78756	3976.00
3	−119.11	5355.7	16.39	140.8947	42,620.00
4	−255.5	11861	36.544	181.0266	94,055.00
5	352.36	−17338	−53.119	NA	NA
6	−35.941	1222	4.032	63.66784	9723.90
7	−44.086	1418.5	5.251	63.68529	11,226.00
8	−133.51	5959.7	18.268	136.2973	47,423.00
9	−65.421	2831.4	8.549	113.3441	22,545.00
10	−118.88	5233.1	16.522	140.9835	41,585.00
11	−81.27	3572.1	10.94	125.9199	28,425.00
12	−54.582	2282.9	7.074	105.5913	18,215.00
13	−73.225	3236.9	9.889	128.3756	25,761.00
14	−203.75	9484.9	29.17	183.3302	75,220.00
15	−80.585	3362.6	11.087	123.8086	26,666.00
16	967.57	−52133	−140.54	NA	NA
17	−95.533	4234.2	12.097	109.262	33,795.00
18	−114.86	5180	15.189	124.6076	41,299.00
19	−141.35	6609	19.113	141.5514	52,723.00
20	−78.114	3418.2	9.746	104.079	27,366.00
21	−31.261	922.3	2.662	43.4634	7402.40
22	−148.99	6728	20.046	131.4074	53,603.00
23	−128.93	5935.5	17.365	136.1121	47,327.00
24	−67.609	2866.9	8.373	98.16466	22,930.00
25	28.875	−2317.9	−5.576	NA	NA

Sl.No*: Serial number and name same as Table 2. ^a^ Newton method is used to calculate the root. N1+N2Tm+N3lnTm=0. ^b^
$\Delta H_{2}^{m}=(-N_{2}+N_{3}T_{m})R$. NA: Not able evaluated the root hence not reported.

## Data Availability

Datas are available from the authors.

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
