# Peer review of "Solubility of Anthraquinone Derivatives in Supercritical Carbon Dioxide: New Correlations"

_molecules, 2021, doi:10.3390/molecules26020460_

Round 1

Reviewer 1 Report

The article entitled: “Solubility of Anthraquinone Derivatives in Supercritical Carbon Dioxide: New Correlationspresent originality on the subject, with high potential in Science area. However, the authors should consider making serious changes mainly related to the drafting of the writing and consider making changes in the text, as:

Title: check line spacing

Authors: include acronyms of each author

Lines 51, 54, 59, 70, 77, 82, 88, 92, 106, 117, 119, 125, 128, 140: include “: at the end of the paragraph prior to the equation

L133. Methodology, include bold word

L155. Include discussions base on the results obtained

L167. Reference: adjust to the format request by the journal

L198. Blank page. Adjust.

Table 3a, b: Authors can use abbreviations (ex NS no significant, S: significant)

L416. References

Adjust base on the authors guide

Update references to no more tan 5 years

L. 562. Include graphic in text

The following sections are not included:

-Author contributions

-Funding

-Conflicts of interest

-Sample availability

Author Response

Dear

Reviewers

We have benefited from the reviewer’s remarks and thank them for their careful analysis and insightful comments. We thank the reviewers for recommending the publication of the paper. We have carefully reviewed the comments by the reviewers and made suitable changes in the manuscript. The remarks of the reviewers are retyped below in italics, our responses are in normal font and the modifications to the manuscript are highlighted in yellow.

Reviewer 2 Report

Solubility of Anthraquinone Derivatives in Supercritical Carbon Dioxide: New Correlations

The work proposes two new models to correlate the solubility of anthraquinone derivatives in supercritical carbon dioxide. In this sense, the authors show different models available in the literature and then present two new models based on solid-liquid equilibrium criteria. The models proposed show useful statistics to correlate this phenomenon. Therefore, I recommend this manuscript to be accepted for publication.

Some commentaries and questions related to this work are:

  1. The article has some small linguistic mistakes. It is recommended to review this.
  2. In lines 37-38, the authors pointed out that the thermodynamic framework based on solid-liquid equilibrium is very successful. However, many models showed in the “Solubility model” section are based on empirical observations. In this sense, just the Chrastil model has a theoretical background. It will be useful to compare the proposed models with another based on SLE.
  3. Equations should be reviewed to uniform nomenclature. Some terms involved are not described in the document body. It will be useful to include them for a better understanding.
  4. Both models (Eq 15 and 17) presented by the authors are based on the fugacity ratio between pure solid and pure liquid. Exploiting the fact that the model is based on SLE criteria, it will be interesting to include analysis, if possible, on the heat of fusion for the different compounds and the corresponding statistical therms computed in the paper. I think this will improve the quality of the manuscript.
  5. Model 1 is presented in a very clear form. However, model 2 is not well understood. The polynomial term (for temperature dependence) is based on the work presented by Nordström and Rasmuson, who fitted the solubility of salicylamide. The authors use this polynomial development for this reason, or is there any other analysis for model 2?
  6. Some authors indicated that there is a linear relationship between pressure and . Is there any reason that the proposed model shows good results, even not including a pressure term?
  7. The authors employed the same reasoning as it is presented by Iwai (1992), Su (2007) to describe the solubility of compounds on supercritical carbon dioxide. These publications used the regular solution model for the activity coefficient. However, the authors use Van Laar; Why?
  8. The authors used van Laar’s model for activity coefficients. They indicated an infinite dilution of solute in the supercritical phase on their theoretical development of equations. However, applying this, activity coefficient should be only determined by . So, is correct (or necessary) the condition of infinite dilution?
  9. In section 4, lines 142 to 146, there are different statistical descriptors. However, in the discussion section, lines 159 to 163, there is just AARD statistical. Maybe the authors should incorporate analysis for other statistical by its significance to compare their model's behavior with the other showed previously.
  10. Table 2 should be reviewed to adjust the number’s presentation.

Author Response

Dear

Reviewers,

We have benefited from the reviewer’s remarks and thank them for their careful analysis and insightful comments. We thank the reviewers for recommending the publication of the paper. We have carefully reviewed the comments by the reviewers and made suitable changes in the manuscript. The remarks of the reviewers are retyped below in italics, our responses are in normal font and the modifications to the manuscript are highlighted in yellow.

Reviewer 3 Report

The authors developed two equations based on the solution model and correlated the solubility data of anthraquinone derivatives in supercritical CO2. This issue is important for process design and development of supercritical dyeing. However, there are several concerns should be discussed before publication. My major concerns are:

  1. In introduction, methods for solubility calculation using different approaches such as equation of state, semi-empirical equation and solution model should be included then illustrate why the solution model was selected. In addition, please also reviewed current situation for solubility calculation of anthraquinone derivatives and point out the motivation of this study.
  2. In the two developed models, the authors should give more discussion for model development. For example, the authors used two empirical equations to represent the fugacity ratio of pure component, and the infinite dilution activity coefficient. Why these empirical equations were used? Frequently,ΔCp term can be neglected in solution model approach. In addition, the authors used van Laar equation to represent the infinite dilution activity coefficient of solute. However, in eq. (14), once the condition is in the infinite dilution, y1 should be approach to 1 and y2 should be closed to zero then Eq. (14) should be equal to A21. Why A12 is still considerable in the final equation for data regression?
  3. Results and discussion section only contain two short paragraphs; the authors should provide more discussion instead of only present the correlation results. In addition, many tables can move to Supporting Information.
  4. Fig. 1 to Fig. 4 compare the calculated and experimental solubilities. However, the curves from two new equations look like discontinuous and not smooth. Furthermore, in Fig. 4, data at 308K showed a minimum value (at CO2 density about 850); data at 318 K showed a considerable solubility decrease at the last point (at highest CO2 density). The new model 2 seems perfectly represent this unusual conditions. Please check the calculation results and discuss.
  5. Please also check all references and cite reference properly.

Author Response

(The authors gave the same response as above.)

Round 2

Reviewer 3 Report

The authors have revised the manuscript and is publishable.